# Real-World Treatment Patterns and Timeliness of Clinical Care Pathway for Non-Small Cell Lung Cancer Patients in Austria: The PRATER Retrospective Study

**DOI:** 10.3390/cancers16142586

**Published:** 2024-07-19

**Authors:** Maximilian Hochmair, Angelika Terbuch, David Lang, Christian Trockenbacher, Florian Augustin, Bahil Ghanim, Dominik Maurer, Hossein Taghizadeh, Christoph Kamhuber, Robert Wurm, Jörg Lindenmann, Petra Braz, Tatjana Bundalo, Merjem Begic, Johanna Bauer, Patrick Reimann, Nino Müser, Florian Huemer, Verena Schlintl, Daniela Bianconi, Bernhard Baumgartner, Peter Schenk, Markus Rauter, Konrad Hötzenecker

**Affiliations:** 1Department of Respiratory and Critical Care Medicine, Karl Landsteiner Institute of Lung Research and Pulmonary Oncology, Klinik Floridsdorf, 1210 Vienna, Austria; 2Division of Oncology, Department of Internal Medicine, Medical University of Graz, 8036 Graz, Austria; 3Department of Pulmonology, Johannes Kepler University Linz, Kepler University Hospital, 4829 Linz, Austria; 4Department of Pulmonology, Klinikum Wels-Grieskirchen, 4600 Wels, Austria; 5Department of Visceral, Transplant and Thoracic Surgery, Medical University Innsbruck, 6020 Innsbruck, Austria; 6Department of General and Thoracic Surgery, University Hospital Krems, 3500 Krems an der Donau, Austria; 7Department of Pulmonology, Ordensklinikum Elisabethinen Linz, 4020 Linz, Austria; 8Division of Oncology, Department of Internal Medicine I, University Hospital St. Pölten, 3100 St. Pölten, Austria; 9Department of Oncology, Kardinal Schwarzenberg Klinikum, 5620 Schwarzach, Austria; 10Department of Pulmonology, Medical University Graz, 8036 Graz, Austria; 11Division of Thoracic and Hyperbaric Surgery, Department of Surgery, Medical University of Graz, 8036 Graz, Austria; 12Department of Pulmonology, Landesklinikum Hochegg, 2840 Hochegg, Austriapeter.schenk@hochegg.lknoe.at (P.S.); 13Department of Thoracic Surgery, Medical University of Vienna, 8036 Vienna, Austria; 14Department of Oncology, Landeskrankenhaus Feldkirch, 6800 Feldkirch, Austria; 15Department of Medicine II with Pneumology, Karl Landsteiner Institute for Lung Research and Pulmonary Oncology, Klinik Ottakring, 1160 Vienna, Austria; 16Division of Pulmonology, Klinik Penzing, 1140 Vienna, Austria; 17MSD, 1100 Vienna, Austria; 18Department of Pulmonology, Vöcklabruck Hospital, 4840 Vöcklabruck, Austria; 19Department of Pulmonology, Klinikum Klagenfurt Am Woerthersee, 9020 Klagenfurt, Austria

**Keywords:** NSCLC, real-world, stage distribution, time intervals, treatment patterns, patient pathway, COVID-19

## Abstract

**Simple Summary:**

Non-small cell lung cancer (NSCLC) is the most common form of lung cancer. Treatments and outcomes for NSCLC are evolving as a result of multiple approvals for immunotherapies and targeted therapies but may also be affected by limitations in clinical care access. In the PRATER study, we described the profile and treatments of patients diagnosed with Stage I–III NSCLC in Austria prior to the introduction of new therapies in the pre-/post-operative setting. We found that therapeutic strategies were aligned with guidelines at that time. Clinical care was timely delivered in most but not all early-stage NSCLC patients. As an additional exploratory objective, we showed that lung cancer care was not significantly affected by COVID-19 restrictions in Austria. Real-world evidence generated here will support future cancer care policies and evaluations of healthcare system efficiency in clinical adoption of new therapies.

**Abstract:**

This was a retrospective study of the profile and initial treatments of adults diagnosed with early-stage (ES) non-small cell lung cancer (NSCLC) during January 2018–December 2021 at 16 leading hospital institutions in Austria, excluding patients enrolled in clinical trials. In total, 319 patients were enrolled at a planned ~1:1:1 ratio across StI:II:III. Most tested biomarkers were programmed death ligand 1 (PD-L1; 58% expressing), Kirsten rat sarcoma virus (KRAS; 22% positive), and epidermal growth factor receptor (EGFR; 18% positive). Of 115/98/106 StI/II/III patients, 82%/85%/36% underwent surgery, followed by systemic therapy in 9%/45%/47% of those [mostly chemotherapy (ChT)]. Unresected treated StIII patients received ChT + radiotherapy [43%; followed by immune checkpoint inhibitors (ICIs) in 39% of those], ICI ± ChT (35%), and ChT-alone/radiotherapy-alone (22%). Treatment was initiated a median (interquartile range) of 24 (7–39) days after histological confirmation, and 55 (38–81) days after first medical visit. Based on exploratory analyses of all patients newly diagnosed with any stage NSCLC during 2018–2021 at 14 of the sites (N = 7846), 22%/10%/25%/43% had StI/II/III/IV. The total number was not significantly different between pre-COVID-19 (2018–2019) and study-specific COVID-19 (2020–2021) periods, while StI proportion increased (21% vs. 23%; *p* = 0.012). Small differences were noted in treatments. In conclusion, treatments were aligned with guideline recommendations at a time which preceded the era of ICIs and targeted therapies in the (neo)adjuvant setting.

## 1. Introduction

Lung cancer accounted for 12% of all newly diagnosed cancers and 19% of all cancer-related deaths worldwide in 2022 [1]. With 5203 incident cases and 4125 deaths in 2022 in Austria, lung cancer accounted for 12% of new cancer diagnoses and 20% of cancer deaths, making it the leading cause of cancer-related deaths [2].

Non-small cell lung cancer (NSCLC) accounts for the majority of lung cancer diagnoses (80–90%) [3,4]. Approximately half of NSCLC patients in Europe [5,6,7,8,9,10], including Austria [11], are diagnosed with stage IV disease. This is important considering that 5-year overall survival (OS) is estimated at ≤10% in metastatic patients [4,12]. In addition, based on literature that precedes the introduction of immune checkpoint inhibitors (ICIs) and targeted therapy (TT) options in the non-metastatic setting [13], both resected and unresected patients with early-stage NSCLC (ES-NSCLC) experience frequent recurrence [14,15,16,17]. The 5-year OS rates range from 92% to 13% across Stage IA1 to IIIC NSCLC [12]. Although prognosis of lung cancer patients in Austria has shown improvements over the years, the last estimated 1-, 3- and 5-year age-standardized net survival was 52% (2015–2017), 30% (2014–2018) and 20% (2010–2015), respectively [18,19]. Furthermore, though trends are not unanimous, some studies in Europe and the US have shown that delays in lung cancer management are associated with poorer prognosis [20,21,22,23]. Altogether these observations highlight the need for improvements in clinical care.

Surgical tumour resection is the principal treatment option for patients with stage I–II NSCLC, followed by adjuvant treatment, if indicated [24,25]. For locally advanced (LA) NSCLC, which encompasses both resectable and unresectable cases, treatment may involve a combination of surgery, systemic therapy (ST) and/or radiotherapy (RT) [24,25]. For patients with metastatic NSCLC, non-surgical ST is the preferred option [26,27]. Platinum-based chemotherapy (ChT) was the standard ST option for first-line (1L) treatment of metastatic NSCLC without actionable oncogenic driver mutations up until European Medicines Agency (EMA)-approval of the first ICI in the 1L setting in 2016 [28,29]. Since then, owing to better treatment outcomes, ICIs (as monotherapy or chemoimmunotherapy) have gradually become a big part of the standard of care for 1L treatment [18,26,28,29]. For oncogene-addicted NSCLC, effective options include TTs, such as tyrosine kinase inhibitors (TKIs) targeting EGFR or ALK [18,27].

The advent of ICIs is also expanding the treatment armamentarium of ES-NSCLC (stage I–III), with first EMA approval in 2018 granted for durvalumab as consolidation therapy for chemoradiotherapy (CRT)-treated unresected stage III NSCLC [30,31]. From 2021 onwards, the indication of multiple pre-existing molecules has been extended to the (neo)adjuvant setting; these include three ICIs for patients with high risk of recurrence [13,32,33,34,35,36,37,38] and one TKI for patients with EGFR-mutated NSCLC [13,39]. Further expansion of indications is awaited [13,40,41,42,43].

As the treatment paradigm for ES-NSCLC is changing, it is imperative to understand real-world (RW) management practices in order to better inform future healthcare decision-making and optimize oncology care, especially in the setting of ES-NSCLC which requires multidisciplinary collaborations. However, Austria lacks a national lung cancer registry [18], while observational studies on local epidemiology are scarce and outdated [11,44]. Thus, the PRATER retrospective study was primarily designed to characterize the profile, therapeutic strategies and journey of patients diagnosed with ES-NSCLC during 2018–2021 using RW data from leading hospital institutions in Austria. Furthermore, in view of the negative impact of the COVID-19 pandemic (2020–2021) on cancer care worldwide [45,46,47,48,49], we collected high-level aggregate data as part of an exploratory objective to capture the potential impact of the pandemic on NSCLC clinical care in Austria.

## 2. Materials and Methods

### 2.1. Study Design and Population

PRATER was a non-interventional, single-country, multicentre, retrospective 2-part study based on medical chart review (Appendix A) carried out at 16 major public hospital clinics.

The primary part focused on individual patient-level data planned to be collected for a sample of approximately 300 patients diagnosed with ES-NSCLC (stage IA–IIIC) between 1 January 2018 and 31 December 2021 (index period) at a 1:1:1: ratio across NSCLC stages I, II and III in order to achieve a meaningful number of patients in each study subpopulation (for more details, see section ‘Statistical methods’). Patients were eligible if diagnosed with histologically or cytologically confirmed ES-NSCLC during the index period; aged ≥18 years at initial NSCLC diagnosis; and with sufficient medical records for data abstraction to meet the study objectives. Patients were selected using consecutive sampling from the end of 2021 backwards, until the target sample was achieved. The aforementioned allocation ratio was monitored through the electronic case report form (eCRF) during the patient accrual period across the study sites. The retrospective data collection period extended from the visit that led to the suspicion of the disease or diagnosis of NSCLC (hereinafter referred to as ‘visit that led to diagnosis’) until the end of the patient’s observation period (EOP); the latter was the earliest date of initial therapeutic strategy (ITS) completion, last contact with the patient, death or study end (31 December 2022). ITS was defined as the sequence of all different treatment modalities (i.e., surgery, RT, ST; excluding supportive treatments) administered immediately after initial diagnosis until earliest date of next treatment initiation or EOP. Data were collected through the eCRF.

In the exploratory part, physicians provided high-level cumulative aggregate information on the number of new NSCLC diagnoses and ITS by year of index period and disease stage, considering all patients diagnosed with NSCLC of any stage (I–IV) at their site during the index period. This information was collected via an electronic survey, in which 14 of the 16 study sites participated. Eligible patients should have been diagnosed with histologically or cytologically confirmed stage I–IV NSCLC during the index period and be ≥18 years of age at initial diagnosis.

Participation in an interventional trial in the context of ITS for NSCLC served as exclusion criterion for the primary part only.

The study conformed to the principles of Declaration of Helsinki, international guidelines for Good Pharmacoepidemiology Practice (GPP) [50], the Strengthening the Reporting of Observational Studies in Epidemiology (STROBE) guidelines [51], the European Union General Data Protection Regulation (GDPR), and local rules and regulations. The protocol was approved by the Institutional Review Boards of the participating sites before any study-related procedures. Informed consent was not required.

### 2.2. Study Objectives

The primary part aimed primarily to characterise the profile of a representative sample of patients diagnosed with ES-NSCLC during the 4-year index period, as well as to shed light on treatment patterns employed as part of ITS, by disease stage at initial diagnosis. Secondary objectives of this part were to capture the diagnostic imaging methods, biomarker testing patterns (from start of diagnostic evaluation until EOP), and the patient journey (from the first point of entry into the healthcare system until the earliest date of ITS completion, last follow-up or death).

The aim of the exploratory part of the study was to record the number and ITS of all patients newly diagnosed with NSCLC at the study sites during the entire index period, and to explore the impact of the COVID-19 pandemic on NSCLC in the country as reflected by variations in NSCLC stage distribution, number of new diagnoses, and changes in treatment modalities during the pre-COVID-19 (2018–2019) and COVID-19 (2020–2021) periods. Further details on the rationale behind the study-specific definition of the COVID-19 period are provided in the Appendix A.

### 2.3. Statistical Methods

A precision-based sample size calculation was employed for the primary study part. The planned size of 300 patients ensured a margin of error < 6% for the study outcomes, with a 95% binomial two-sided confidence level using normal approximation within 4.4–5.6%. This size also ensured a margin of error < 10% for the frequency estimates in the subpopulations by disease stage comprising at least 96 patients each.

Categorical variables are presented as frequencies and continuous variables as mean and standard deviation (SD) or median and interquartile range (IQR; reported as Q_1_–Q_3_) depending on the normality of data based on the Shapiro-Wilk test. No imputation methods were applied except for partial dates.

Percent change in the number of NSCLC diagnoses between COVID-19 and pre-COVID-19 periods were calculated as the difference between the total number of patients diagnosed in 2020–2021 and those diagnosed in 2018–2019, then divided by the latter and multiplied by 100. Statistical significance of difference between number of diagnoses as well as ST frequencies was examined using the chi-squared (X2) or Fisher’s exact test.

All statistical tests were two-sided and performed at a 0.05 significance level. Sample size determination and statistical analyses were performed using SAS v9.4 (SAS Institute, Cary, NC, USA).

## 3. Results

### 3.1. Primary Part: Individual Patient-Level Data for Patients Initially Diagnosed with Stage I–III NSCLC

#### 3.1.1. Patient Disposition

From 21 February 2023 to 24 March 2023, a total of 323 patients were included from 16 hospital clinics in Austria. Four patients did not meet all eligibility criteria, thus the analysis included 319 patients.

Patient distribution was balanced between non-academic and academic institutions, in terms of location, 68.3% were included by sites outside Vienna (Appendix A). Approximately half of patients (55.5%; 177/319) were under the care of the study site at the ‘visit that led to diagnosis’. Median (IQR) time from initial NSCLC diagnosis (i.e., date of histological confirmation) until EOP was 2.7 (1.1–5.7) months; EOP was triggered by the ‘end of ITS’ in the majority of patients (79.6%; 254/319) (Appendix A). At EOP, 91.9% (274/298) of patients with available data (evaluable population) had the same disease stage as initially diagnosed (Appendix A).

As per study design, patients were evenly distributed among Stage I, II and III NSCLC at initial diagnosis [36.1% (115/319), 30.7% (98/319), and 33.2% (106/319), respectively]. Patient disposition details within each stage are presented in Appendix A.

#### 3.1.2. Patient Characteristics

Patient sociodemographic and clinical characteristics at initial ES-NSCLC diagnosis, overall and per stage are shown in Table 1. Initial diagnostic assessment involved imaging in 97.2% (310/319) of the overall population (Table 1). Disease staging was based on AJCC/UICC TNM 8th Edition criteria in 99.0% (284/287) of evaluable patients.

Overall, 94.0% (300/319) of patients underwent biomarker testing during the retrospective observation period, performed a median of 0.0 (0.0–3.5) days from diagnosis. For 51.7% (155/300) of the tested patients a commercially available panel was used, for 27.7% (83/300) specific biomarkers were examined, while 20.7% (62/300) were tested using both commercially available panel and specific biomarker testing.

Testing methodology and findings for the most frequently tested biomarkers among patients who underwent specific biomarker testing are presented in Figure 1 and Appendix A. The highest positivity rates were observed for PD-L1 (TPS ≥ 1% via immunohistochemistry; 57.9%) (Figure 1A), KRAS (22.2%), and EGFR (18.2%) (Figure 1B). Slight variations were observed between subpopulations by stage, with Stage I having the lowest PD-L1 and highest EGFR positivity rates across stages, whereas the opposite was noted for Stage II.

#### 3.1.3. Initial Treatments

The majority of the overall early-stage NSCLC population (96.2%; 307/319) received (non)pharmacologic treatment as part of ITS (Table 2).

Most Stage I–II patients (83.1%; 177/213) and 35.8% of Stage III patients underwent surgery (Table 2). ST rates increased with advancing disease stage from 7.8% in Stage I to 43.9% in Stage II and 82.1% in Stage III. ST rates mostly reflect adjuvant ST post-surgery in Stage I (7.0%; 8/115) and Stage II patients (37.8%; 37/98). Conversely, ST rate only partially reflects adjuvant ST in Stage III patients (17.0%; 18/106) since 54.7% of Stage III patients were unresected and treated with ST ± RT. More than a third of Stage III patients (35.8%) were treated with RT (±ST/surgery), which was CRT in 81.6% (31/38) of cases (Table 2).

Among treated patients in the Stage I, II and III subpopulations, 7.0% (8/114), 44.2% (42/95), 61.2% (60/98) were managed with a multimodal approach. Most common management strategies (frequency ≥ 10.0%) were: ‘surgery-alone’ in 75.4% (86/114) and ‘RT-alone’ in 16.7% (19/114) of the treated Stage I; ‘surgery-alone’ in 43.2% (41/95), and ‘surgery + ST’ in 38.9% (37/95) of treated Stage II; ‘ST + RT’ in 30.6% (30/98), ‘ST-alone’ in 28.6% (28/98), and ‘surgery + ST’ in 24.5% (24/98) of treated Stage III patients.

Treatment sequences, including ST pharmacologic categories and drug classes, for each disease stage are depicted in Figure 2.

Platinum-based ChT was the most frequent ST administered across stages, while ICI and TT was reported for 28.1% and 0.7%, respectively, the latter two mainly driven by the Stage III subpopulation (Table 2). Unresected Stage III patients were treated with ChT + RT (43.3%; 26/60), ICI ± ChT (±RT; 35.0%; 21/60), ChT-alone (18.3%; 11/60), or RT-alone (3.3%; 2/60); ChT + RT was followed by ICI (anti-PD-L1) in 16.7% (10/60). ST among resected Stage III patients comprised ChT in 57.9% (22/38), ICI ± ChT in 15.8% (6/38), and TT in one patient (2.6%) (Figure 2).

Further details on surgical therapies, reasons for not performing surgery and not administering ST as well as timing of therapy are provided in Table 2.

#### 3.1.4. Patient Journey

The most common reason for the ‘visit that led to diagnosis’ was ‘incidental medical findings’ (51.7%; 165/319), followed by ‘symptoms’ (37.3%; 119/319), ‘lung cancer screening’ (14.7%; 47/319) and ‘during admission in the emergency room’ (2.5%; 8/319) (Appendix A). Pulmonologists and primary care physicians were the most frequent specialties involved in this visit [53.6% (158/295) and 10.8% (32/295) of evaluable patients, respectively].

Among patients who were not under the medical care of the study site during the ‘visit that led to diagnosis’, (N = 142), the majority were either under the care of another public peripheral hospital, or the physician’s own public practice (Appendix A). These patients first visited the study site after a median (IQR) of 28.5 (12.0–70.0) days (Appendix A).

Imaging testing was performed a median (IQR) of 0.0 (−11.0 to 3.0) days from the ‘visit that led to diagnosis’ (Figure 3). In the overall population, i.e., including patients who did not undergo imaging testing (N = 319), NSCLC was histologically confirmed a median (IQR) of 25.0 (10.0–52.0) days from the ‘visit that led to diagnosis.’ Among treated patients (N = 307), ITS was initiated a median of 24.0 days after histological confirmation, for a total median time of 55.0 days from the ‘visit that led to diagnosis’ (Figure 3).

In terms of treatment decision-making, 98.7% (312/316) of evaluable patients were presented in the corresponding multidisciplinary tumor board. In more than half of patients, patient care during ITS was followed-up by pulmonologists (Appendix A).

Timeliness of patient journey by reason and healthcare setting is presented in Figure 3, and by gender and health insurance in Appendix A. Some differences were noted in a purely descriptive manner. Specifically, the time taken to start ITS was slightly longer for patients: with incidentally detected NSCLC; who were not under the care of the study site from the beginning of their NSCLC journey; and, who did not have additional private health insurance.

### 3.2. Exploratory Part: Aggregate Data for Patients Initially Diagnosed with Stage I–IV NSCLC

During the entire 4-year index period (2018–2021), 7846 patients were newly diagnosed with NSCLC at any stage at the 14 participating sites. The majority of patients (67.9%; 5326/7846) were under the care of hospital sites located outside Vienna, while 56.4% (4429/7846) received care by non-academic institutions.

An even distribution of NSCLC diagnoses was observed throughout the years, with 24.5% (1921/7846), 24.8% (1949/7846), 24.0% (1883/7846) and 26.7% (2093/7846) being diagnosed in years 2018, 2019, 2020 and 2021, respectively. The number of new NSCLC diagnoses at the study sites did not significantly differ (*p* = 0.231) between the pre-COVID-19 and COVID-19 periods (3870 and 3976 diagnoses, respectively; representing a relative increase of 2.7%).

Throughout the index period, most frequent disease stage at diagnosis was Stage IV (43.3%), followed by Stage III (24.9%), Stage I (22.0%) and Stage II (9.9%) (Figure 4A). Patient distribution by institution location and type is presented in Appendix A. The number of patients diagnosed with NSCLC across the different stages per year is depicted in Figure 4B. The distribution of new Stage II, III and IV NSCLC diagnoses changed slightly but did not statistically significantly differ between the pre-COVID-19 and COVID-19 periods, whereas the proportion of Stage I patients significantly increased during the pandemic (n = 920; 23.1%) compared to the pre-pandemic period (n = 805; 20.8%) (*p* = 0.012; increase by 14.3%) (Figure 4C).

Overall, 2.3% (40/1725), 3.2% (25/775), 5.5% (108/1950), and 12.9% (439/3396) of Stage I, II, III and IV patients, respectively, did not receive any therapy as part of ITS.

Among Stage I–II NSCLC patients, 78.4% (1959/2500) underwent surgery and 19.5% (487/2500) received RT (without surgery), as part of ITS. Surgery was coupled with (neo)adjuvant RT or ST in a higher percentage of resected Stage II (54.2%; 325/600) than Stage I patients (6.6%; 90/1359). Among the unresected RT-treated patients, ST was administered in a slightly higher percentage of Stage II (34.7%; 43/124) than Stage I patients (23.7%; 86/363). ST-alone was administered in 3.8% (96/2500) of Stage I–II patients; of the latter, 45.8% (44/96) received investigational ST, 24.0% (23/96) ICI ± ChT, 22.9% (22/96) ChT-alone, and 7.3% (7/96) TT ± ChT.

Compared with earlier stages, surgery rates were notably lower among Stage III NSCLC patients (40.6%; 792/1950), while adjuvant treatment rates among resected patients were higher: 78.9% (625/792) in Stage III versus 5.2% (71/1359) in Stage I and 46.7% (280/600) in Stage II. Adjuvant treatment among resected Stage III patients comprised (C)RT in 78.2% (489/625) of cases, ICI in 18.2% (114/625), and TT in 3.5% (22/625). RT (without surgery) was more frequent among Stage III NSCLC patients (28.4%; 554/1950) compared with earlier stages, and was part of CRT in 77.8% (431/554) of these cases. A fourth (25.6%; 499/1950) of Stage III NSCLC patients received ST-alone, mainly comprising ChT-alone (36.1%; 180/499) or ICI ± ChT (33.5%; 167/499).

Among Stage IV patients, 10.3% (351/3396) received non-pharmacologic treatment only, and 76.7% (2606/3396) received ST. Of the latter, 61.9% (1614/2606) received ICI (± ChT and/or TT), 17.9% (467/2606) ChT-alone, 14.9% (387/2606) TT (± ChT), and 5.3% (138/2606) investigational ST.

Treatments utilized as part of ITS in the pre-COVID-19 and COVID-19 periods in the Stage I/II/III/IV NSCLC subpopulations are illustrated in Figure 4D. For those receiving ST-alone, the ST rate was statistically significantly lower in the pandemic versus pre-pandemic period among Stage I NSCLC [1.4% (13/920) vs. 3.0% (24/805); *p* = 0.025]. The opposite was observed for Stage IV patients with statistically significantly higher ST rates in the pandemic versus pre-pandemic period [78.5% (1352/1722) vs. 74.9% (1254/1674); *p* = 0.013]. ST rate did not statistically differ between the aforementioned periods for Stage II and III patients. In terms of ST pharmacologic categories, the frequency did not statistically significantly differ between two periods, neither for Stage I nor for Stage II patients. Conversely, a statistically significant decrease was noted in the rate of ChT-alone and a statistically significant increase in the rate of ICI ± ChT, for both Stage III and IV subpopulations (Figure 4D).

### 3.3. Alignment between Primary and Exploratory Findings

ITS patterns were collected at a patient level for the primary part whereas data were collected in aggregate form via an electronic survey for the exploratory part. In view of these differences, the concordance in findings between the two study parts is presented below, albeit in a purely descriptive manner.

Rates of surgery, postoperative/adjuvant ST ± RT and RT-alone were similar between the primary and exploratory part of the PRATER study for both Stage I [surgery (81.7% vs. 78.8%); postoperative/adjuvant ST ± RT (7.0% vs. 4.1%); RT without surgery (16.5% vs. 21.0%)], and Stage II patients [surgery (84.7% vs. 77.4%); postoperative ST ± RT (37.8% vs. 36.1%); RT without surgery (9.2% vs. 16.0%)].

Among Stage III patients, surgery rates were similar between the primary and exploratory parts (35.8% vs. 40.6%). In the primary part, adjuvant ICI ± ChT and TT was reported in 2.8% (3/106) and 0.9% (1/106) of Stage III patients, respectively (Figure 2). Rates of adjuvant ICI and TT among Stage III patients in the exploratory part were 5.8% (114/1950) and 1.1% (22/1950), suggesting higher ICI utilization. Rate of ST and/or RT without surgery was consistent between the two study parts: in the primary part, ST ± RT was 54.7%, while in the exploratory part, ST-alone was 25.6% and CRT (±other ST) was 22.1%. Lastly, ChT-alone (±RT; without surgery) was similar: in the primary part, ChT-alone (±RT) was 25.5% (27/106), while in the exploratory part, ChT-alone was 9.2% (180/1950) and CRT-alone was 12.7% (248/1950).

## 4. Discussion

### 4.1. Primary Part

The primary part of PRATER captured RW characteristics and ITS of patients diagnosed with ES-NSCLC in Austria during 2018–2021, a period which precedes the era of (neo)adjuvant ICI/TT [13,24,25,26,27,28,30,31].

A comprehensive picture of the local RW patient population is provided, which differs from clinical trial populations that are often confined by strict and narrow eligibility criteria. This information is valuable for identifying appropriate candidates for newly approved and upcoming biomarker-guided treatments. PD-L1-positivity TPS ≥ 50%/≥ 1% was 24%/58%, which is similar to previously reported European rates: 20–30%/49–63% varying stage [10,52,53,54,55,56,57,58]; 17–18%/46–54% Stage I–IIIA [53,57]; and 21–27%/49–59% Stage IIIB–IV [52,53,55,56,57]. EGFR-positivity was 18% in PRATER which is within the European range: 6–22% varying stage [10,44,52,53,55,56,57,58,59,60,61,62,63,64,65,66]; 10–20% Stage I–III [44,53,57,59,67]. Although some variations were noted between stages in PRATER, the absence of statistical assessments and the small sample sizes do not allow for meaningful inferences. Literature in this respect is still limited and conflicting [53,57,59,64,67]. Nevertheless, our findings provide preliminary evidence of the sizeable fraction of ES-NSCLC that would potentially benefit from biomarker-guided therapy. Furthermore, 94% of PRATER ES-NSCLC patients had undergone (any) biomarker testing, higher than prior RW European rates (26–75%) [44,57] in the pre-ICI/TT era of ES-NSCLC. As expected, PD-L1 and EGFR were among the most commonly tested biomarkers. With upfront reflex testing becoming common practice [18,68,69] and an ongoing search for reliable predictors of neoadjuvant response [70], testing rates are expected to further increase, as previously documented [55,71]. Biomarker prevalence and associated outcomes warrant future investigations to fully understand the population impact of genetics-guided lung cancer care.

In PRATER, ITS varied across stages and was in agreement with guidelines at that time [3,13,24,30,31]. In summary, Stage I patients were mostly treated with ‘surgery-alone’ (75%), and the majority of Stage II patients underwent surgery (85%) which was combined with pre-/post-operative ST in nearly half of these cases (48%). The predominant ITS among Stage III patients was ST ± RT (55%), while the rest mostly underwent surgery (36%) along with pre-/post-operative ST in 76% of these cases. This trend in ST rates is also aligned with a higher (neo)adjuvant ChT use with advancing stage previously documented [72]. RT rates in PRATER (Stage I–II: 15%; III: 36%) were lower, whereas surgery rates (Stage I–II: 85%; III: 36%) were higher than previously reported in Europe (years 2010–2019); 22–44% RT and 37–68% surgery in Stage 0–II [5,7,8,73,74,75]; >40–62% RT (2–26% RT-alone) and <26% surgery in Stage III [5,7,8,44,54,73,75,76]. In terms of pharmacologic category, adjuvant therapy in PRATER comprised ChT in all Stage I–II cases. Most ST was ChT among resected Stage III patients, while 78% (47/60) of unresected patients were treated with ChT + RT (± subsequent ICI) or ICI ± ChT which are the preferred guideline-recommended options for unresected Stage IIIA/IIIB and Stage IIIB/IV patients, respectively [24,25,26].

Altogether the above findings, indicate that the participating sites closely adhere to evidence-based recommendations. Nevertheless, a non-negligible proportion of ES-NSCLC patients (12%) did not receive standard initial therapy [i.e., ST-alone for Stage I–II (n = 4); RT-alone for Stage II (n = 9); ChT-alone/RT-alone (n = 13) for Stage III; no treatment at all (n = 12)] illustrating continued unmet NSCLC therapeutic needs. Data presented herein will serve as a benchmark for assessing the optimal uptake of new treatments and delivery of quality healthcare.

In the primary part of the study, timeliness of healthcare for patients with early-stage NSCLC was also examined. Differences in study designs, time interval definitions, periods examined and other factors such as NSCLC stage (late-stage cancer patients receive treatment faster) [21,23] or patient behavior, render it challenging to discuss patient journey in relation to literature [20,21,22,23,77]. To provide, however, a context in which to interpret the results, PRATER showed that, median time from entry into the Austrian healthcare system (i.e., ‘visit that led to diagnosis’) until 1st visit at the study site was relatively long (29 days) when considering relevant literature: median time from primary care visit/1st specialist referral until 1st specialist appointment ranged 1–20 days [20,22,23,77]. Conversely, median time from entry into the healthcare system until diagnosis (25 days), and from diagnosis to ITS start (24 days), were aligned with previously reported European range of medians: 28–65 [77], and 6–45 [11,22,23,77] days, respectively. Still, long waiting times were reported for a significant proportion of patients; 25% of patients (75th quartile) reached the study site >70 days after entry into the system, received their NSCLC diagnosis >50 days after imaging, started treatment >39 days after diagnosis, and started treatment a total of >81 days after entry into the system. These intervals exceed the UK-recommended 62-day target (between date of cancer suspicion and treatment initiation) [78], suggesting there is room for improvement in the timeliness of the entire care trajectory which is critical for patients’ faster access to therapy and potentially better prognosis.

### 4.2. Exploratory Part

The exploratory part of the present study provides RW evidence from a large generalizable pool of 7846 patients newly-diagnosed with NSCLC during 2018–2021. The percentage of patients with Stage I, II, III and IV was 22%, 10%, 25% and 43%, respectively, in line with ranges reported across European countries: 12–24% [5,6,7,8,10,74], 6–10% [5,6,7,8,10,74], 19–26% [5,6,7,8,10], and 48–55% [5,6,7,8,10]. Only small differences were noted compared with earlier Austrian 2013–2015 data [11]. Specifically, Stage I–II NSCLC accounted for slightly more NSCLC cases in PRATER (32% vs. 24%), which might reflect advancements in patient awareness and healthcare provision or other epidemiological trends. Potential implementation of a lung screening program may lead to further shifts in stage distribution in the future [18].

The COVID-19 pandemic had a significant impact on healthcare systems worldwide, including referral, diagnosis and management of lung cancer [45,46,47,48,49]. International literature not only reported a drop in the number of newly-diagnosed NSCLCs during the pandemic, but also an increase in tumour stage, suggesting that pandemic-imposed restrictions and delays led to more patients being diagnosed at more advanced stages [45,46,47,48,49,79]. The findings of the exploratory part of PRATER were not aligned with literature trends. The number of new NSCLC cases remained relatively stable, while in terms of staging, a significant increase in Stage I NSCLC was observed. The latter observation could be speculatively partially attributed to an increase in chest CT imaging taking place during the pandemic for suspected or documented COVID-19 infection [80,81,82].

When assessing the impact of COVID-19 in cancer care, diagnosis and treatments should both be considered. Although use of ST-alone among Stage I patients in PRATER was statistically significantly reduced during the pandemic, this difference is not considered of clinical importance given the small proportion of patients belonging to this treatment group. Furthermore, based on literature, a decline in surgical interventions was observed during the pandemic [47,48]. This was not the case in PRATER. Lastly, the statistically significant increase in ICI among Stage III–IV patients, coupled with a statistically significant decrease in ChT-alone, reflects the continued approval and adoption of new ICIs, rather than a pandemic-related cause [18,26,28,29,30,31]. Consistently, higher 1L ICI usage among Stage IV patients during COVID-19 has previously been documented in other European countries as well, most likely arising from the evolving treatment landscape as stated by the authors [83].

The overall ST rate (79%) and ICI rate (53%) among Stage IV patients observed during the pandemic in PRATER closely follows 2020 ESMO-based benchmarks for Stage IIIB/IV of ~75% and ~40%, respectively (based on guidelines and timing of EMA approval) [29]. ICIs are becoming the cornerstone 1L therapy for non-oncogene addicted Stage IIIB–IV patients, while ChT-alone is reserved as an option for patients with low/absent PD-L1 expression, poor performance status, and ICI contraindications, among other clinical factors [26,83,84]. As such, the low utilization of ChT-alone among metastatic NSCLC patients (9%) during the pandemic in PRATER indicates that treatment selection is very much aligned with temporal trends, more so than another EU5 study (37% of 1L regimens in Stage IV were ChT-alone) [83]. In addition to optimal treatment decision-making, compulsory social insurance [85], and EMA-based centralized authorization [86,87] allowing fast access to breakthrough therapies, have likely contributed to the observed Austrian trends.

Overall, lung cancer care in Austria was not significantly affected by the COVID-19 pandemic in healthcare encounters, while new pharmacologic therapies are being successfully implemented in clinical practice.

### 4.3. Strengths and Limitations

The primary strength of PRATER study is the capture of RW data from leading hospital institutions that treat NSCLC in Austria using robust datasets. The sample of 319 patients of the primary study part met the originally planned size. A large population was analyzed for the exploratory part: 7846 patients representing ~46% of all NSCLC diagnoses during 2018–2021 in Austria (estimating ~17,000 cases based on an average ~5000 newly diagnosed lung cancer cases annually and ~85% being NSCLC). Furthermore, a close alignment of treatment patterns was observed between the two study parts, thereby lending credibility to the study findings.

As with any retrospective study, PRATER bears several limitations, including patient selection, confounding and information bias. Efforts were made to minimize such bias by applying consecutive sampling and enrolling patients with sufficient medical records (applicable for the primary part only). Furthermore, stage distribution could be biased due to non-probability sampling for study site selection (e.g., three thoracic surgery departments were included). Additionally, although disease staging in the primary part was mostly based on AJCC/UICC TNM 8th Edition criteria, this information is lacking for the exploratory part due to the data being collected in a cumulative high-level aggregate form, as per study design. Therefore, interpretations of the exploratory part results should take into consideration that differences in staging system may exist across hospital sites.

Regarding biomarker testing, no specific central laboratory testing was enforced, thus the possibility of inherent inter- and intra-assay/laboratory/observer bias exists. Another source of information bias arises from the fact that, for commercially available panels, the specific biomarkers that tested negative were not collected/recorded, thus relevant data were excluded from the analysis.

Lastly, the study was not designed to statistically analyze differences between subpopulations, thus any observations are presented in a purely descriptive approach. Any interpretation of treatment patterns should also take into account that differences may exist between substages which were not captured herein. Future substage analyses are warranted in light of the new ICI/TT indications in the non-metastatic setting.

## 5. Conclusions

In Austria in 2018–2021, the majority of ES-NSCLC patients were treated with guideline-recommended therapies and received timely medical care. However, a small proportion still did not receive standard therapies and experienced long intervals between critical touchpoints along their healthcare journey, altogether indicating there is still room for improvement. Furthermore, the study showed that in Austria, two thirds (68%) of NSCLC patients are diagnosed at advanced stages (III/IV). COVID-19 restriction measures did not significantly affect clinical care of NSCLC patients, as evidenced by the number of new NSCLC diagnoses, stage distribution, and therapies utilized, which remained largely unaffected. ICIs were successfully implemented despite the pandemic, demonstrating the efficiency of the healthcare system. Evidence generated in the PRATER study will be valuable in planning future lung cancer policies.

## Figures and Tables

**Figure 1 cancers-16-02586-f001:**
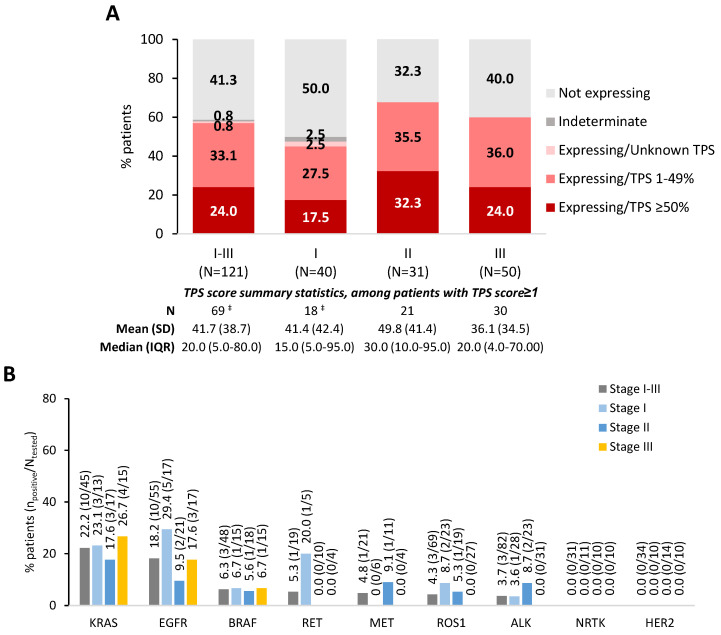
Findings for biomarkers, among patients with early-stage NSCLC who underwent specific biomarker testings ^†^, from start of diagnostic evaluation of lung cancer until EOP, overall and per NSCLC stage at initial diagnosis: (**A**) PD-L1 TPS via IHC, and (**B**) other biomarkers examined in ≥20.0% of patients in any subpopulation. ^†^ Excluding patients tested using commercially available panels, since for those patients specific biomarkers that tested negative were not recorded, thus introducing information bias in the respective positivity rates. ^‡^ Specific TPS score was missing for 1 patient with Stage I NSCLC. Abbreviations: ALK, Anaplastic Lymphoma Kinase; EGFR, Epidermal Growth Factor Receptor; EOP, End of Observation Period; HER2, Human Epidermal Growth Factor Receptor 2; IHC, Immunohistochemistry; IQR, Interquartile Range; KRAS, Kirsten Rat Sarcoma Virus; N, number of patients with available data (i.e., tested for specific biomarker); NRTK, Neurotrophic Receptor Tyrosine Kinase; NSCLC, Non-Small Cell Lung Cancer; PD-L1, Programmed Death Ligand 1; SD, Standard Deviation; TPS, Tumour Proportion Score.

**Figure 2 cancers-16-02586-f002:**
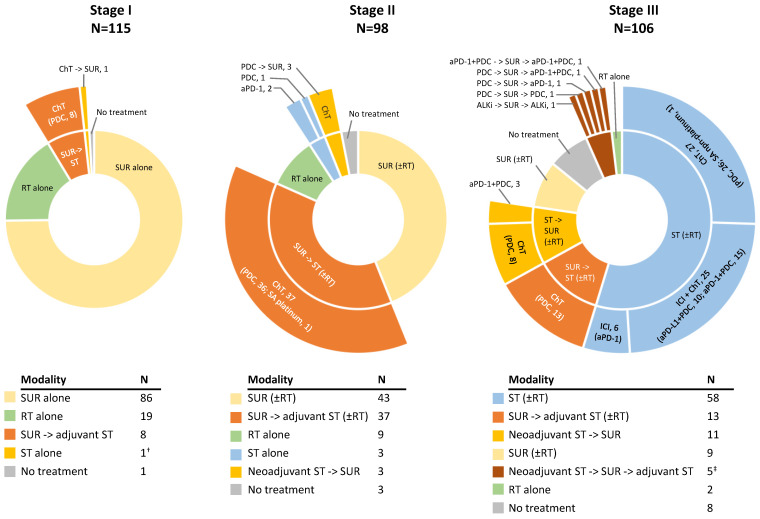
Results based on individual patient-level data for patients initially diagnosed with stage I–III NSCLC: combinations of treatments received as part of initial therapeutic strategy for early-stage NSCLC, per disease stage at initial diagnosis. For each NSCLC stage, inner pie chart shows modality (i.e., higher level) and outer pie chart shows pharmacologic category and drug class (i.e., lower level). ^†^ Recorded as neoadjuvant ST but did not undergo surgery. ^‡^ One Stage III patient received ChT spanning both the pre- and post-operative periods, thus ST was included as adjuvant therapy as well. Abbreviations: ALKi, Anaplastic Lymphoma Kinase inhibitor; aPD-1, anti-Programmed Cell Death 1; aPD-L1, anti-Programmed Death Ligand 1; ChT, Chemotherapy; ICI, Immune Checkpoint Inhibitor; N, number of patients with available data; NSCLC, Non-Small Cell Lung Cancer; PDC, Platinum Doublet Chemotherapy; RT, Radiotherapy; SA, single-agent; ST, Systemic Therapy, SUR, Surgery.

**Figure 3 cancers-16-02586-f003:**
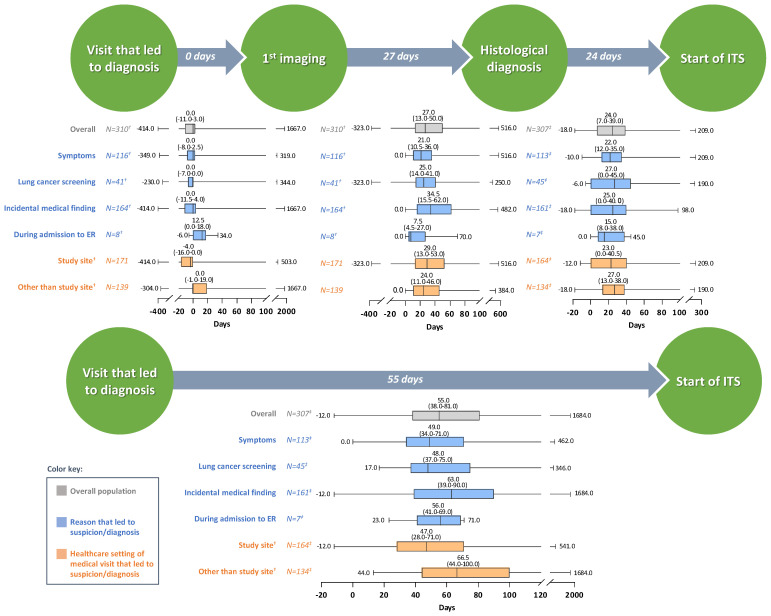
Time from the visit that led to the suspicion of the disease or NSCLC diagnosis until start of initial therapeutic strategy for early-stage NSCLC, per reason that led to suspicion/diagnosis and healthcare setting of the medical visit that led to suspicion/diagnosis. Numbers inside arrows indicate median time for the overall population. Box-plots depict median with interquartile range. ^†^ Among patients who underwent diagnostic imaging testing. ^‡^ Among patients that received any treatment as part of ITS. Abbreviations: ER, Emergency Room; ITS, Initial Therapeutic Strategy; N, number of patients with available data; NSCLC, Non-Small Cell Lung Cancer.

**Figure 4 cancers-16-02586-f004:**
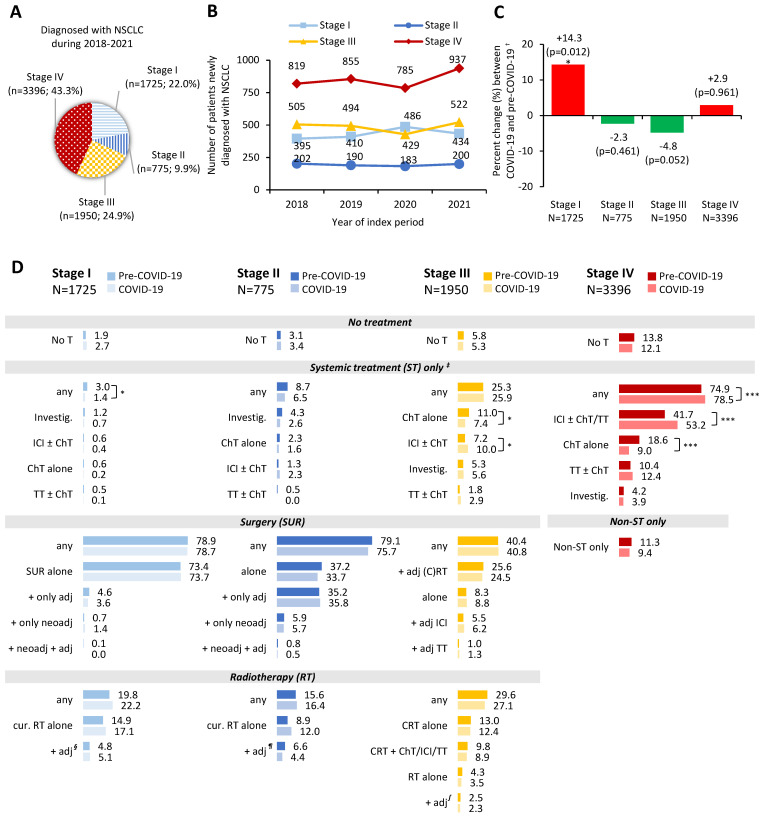
Results based on aggregate high-level data for patients initially diagnosed with stage I–IV NSCLC: (**A**) NSCLC stage distribution; (**B**) Number of patients newly diagnosed with NSCLC per stage and year of diagnosis; (**C**) percent change in number of patients newly diagnosed with NSCLC between COVID-19 (2020/21) and pre-COVID-19 (2018/19) periods; and (**D**) Type of initial therapeutic strategy, per NSCLC stage at initial diagnosis. ^†^ Asterisk (*) indicates statistically significant difference of *p* ≤ 0.05. ^‡^ Statistical differences are shown with asterisks: * *p* ≤ 0.05, *** *p* ≤ 0.001. ^§^ Any adjuvant treatment (including ChT, TT, ICI alone or in combination). ^¶^ Any additional treatment (including concurrent or sequential CRT). ^∫^ Any adjuvant treatment (including TT, ICI or TT + ICI). Abbreviations: adj., adjuvant; ChT, Chemotherapy; CRT, Chemoradiotherapy; cur., curative; Investig., Investigational; ICI, Immune Checkpoint Inhibitor; N, number of patients with available data; n, number of patients with variable; neoadj., neoadjuvant; NSCLC, Non-Small Cell Lung Cancer; RT, Radiotherapy; ST, Systemic Therapy, SUR, Surgery; T, Therapy; TT, Targeted Therapy.

**Table 1 cancers-16-02586-t001:** Patient sociodemographic and clinical characteristics at initial diagnosis of early-stage NSCLC ^†^, overall and per disease stage at initial diagnosis.

		Overall (N = 319)	Stage I (N = 115)	Stage II (N = 98)	Stage III (N = 106)
Age (years)	median (IQR)	69.0 (62.0–76.0)	70.0 (62.0–76.0)	65.5 (61.0–74.0)	70.0 (60.0–76.0)
	<70, % (n/N)	51.7 (165/319)	49.6 (57/115)	58.2 (57/98)	48.1 (51/106)
Male, % (n/N)		53.3 (170/319)	44.3 (51/115)	53.1 (52/98)	63.2 (67/106)
Ever smokers (current & former), ^‡^ % (n/N)		92.8 (270/291)	88.3 (91/103)	95.6 (86/90)	94.9 (93/98)
Pack-years of smoking, median (IQR)		40.0 (29.3–50.0)	36.3 (20.5–50.0)	40.0 (30.0–50.0)	45.0 (30.0–52.5)
BMI (kg/m^2^), ^‡^ median (IQR)		25.1 (22.3–28.4)	26.0 (22.5–29.7)	25.9 (23.4–29.0)	24.0 (21.5–26.6)
ECOG performance status score 0–1, ^‡^ % (n/N)		89.3 (201/225)	92.9 (79/85)	90.8 (59/65)	84.0 (63/75)
History of other primary malignancies, % (n/N)		21.6 (69/319)	25.2 (29/115)	20.4 (20/98)	18.9 (20/106)
Histological disease diagnosis, % (n/N)		94.4 (301/319)	98.3 (113/115)	91.8 (90/98)	92.5 (98/106)
Primary tumor size (cm), ^‡^ median (IQR)		2.9 (1.8–4.7)	1.9 (1.3–2.7)	3.8 (2.4–4.8)	4.7 (2.5–7.2)
Most common primary tumor histology, ^‡,⌠^ % (n/N)	Non-squamous	50.6 (159/314)	63.2 (72/114)	47.4 (46/97)	39.8 (41/103)
	Squamous	35.4 (111/314)	21.1 (24/114)	38.1 (37/97)	48.5 (50/103)
	Adenosquamous	11.5 (36/314)	11.4 (13/114)	12.4 (12/97)	10.7 (11/103)
Imaging testing performed, % (n/N)		97.2 (310/319)	96.5 (111/115)	98.0 (96/98)	97.2 (103/106)
Most common ^∫^ type of testing performed, % (n/N)	CT	92.9 (288/310)	95.5 (106/111)	95.8 (92/96)	87.4 (90/103)
	PET-CT	53.9 (167/310)	55.9 (62/111)	55.2 (53/96)	50.5 (52/103)
	Brain MRI	25.5 (79/310)	24.3 (27/111)	32.3 (31/96)	20.4 (21/103)

For variables not following a normal distribution in at least one of the study subpopulations, a uniform presentation of median (IQR) was applied. ^†^ Date of confirmation of initial NSCLC diagnosis or within the preceding 30 days; ^‡^ Data was missing/unknown in a subset of patients for ‘smoking status’ (n = 28), ‘ECOG performance status’ (n = 94), ‘BMI’ (n = 34), ‘primary tumor size’ (n = 63), ‘primary tumor histology’ (n = 5), and ‘type of imaging testing’ (n = 9). ^⌠^ Six more subtypes were reported in ≤2 patients each. ^∫^ Reported in ≥10.0% in any of the examined subpopulations. Abbreviations: BMI, Body Mass Index; CT, Computed Tomography; ECOG, Eastern Cooperative Oncology Group; IQR, Interquartile Range; MRI, Magnetic Resonance Imaging; N, number of patients with available data; n, number of patients with variable; NSCLC, Non-Small Cell Lung Cancer; PET-CT, Positron Emission Tomography–Computed Tomography.

**Table 2 cancers-16-02586-t002:** Treatment modalities received as part of initial therapeutic strategy for early-stage NSCLC, overall and per disease stage at initial diagnosis.

	Overall (N = 319)	Stage I (N = 115)	Stage II (N = 98)	Stage III (N = 106)
**Frequencies of treatment modalities, % (n/N)**				
Any (non)pharmacologic treatment (SUR, ST, and/or RT) ^†^	96.2 (307/319)	99.1 (114/115)	96.9 (95/98)	92.5 (98/106)
SUR (±RT or ST)	67.4 (215/319)	81.7 (94/115)	84.7 (83/98)	35.8 (38/106)
Receipt of (neo)adjuvant ST, among resected patients				
Receipt of neoadjuvant ST	8.8 (19/215)	.	3.6 (3/83)	42.1 (16/38)
Receipt of adjuvant ST	29.3 (63/215)	8.5 (8/94)	44.6 (37/83)	47.4 (18/38)
Type of resection, among resected patients ^⌠^				
Lobectomy	77.7 (167/215)	76.6 (72/94)	81.9 (68/83)	71.1 (27/38)
Segmentectomy	7.9 (17/215)	13.8 (13/94)	3.6 (3/83)	2.6 (1/38)
Wedge resection	7.4 (16/215)	9.6 (9/94)	6.0 (5/83)	5.3 (2/38)
Surgical approach, among resected patients (unknown for n = 4) ^∫^				
VATS	50.7 (107/211)	71.0 (66/93)	40.2 (33/82)	22.2 (8/36)
Thoracotomy	46.9 (99/211)	24.7 (23/93)	58.5 (48/82)	77.8 (28/36)
Surgical Margin, among resected patients (unknown for n = 12)				
R0	95.6 (194/203)	97.8 (90/92)	93.4 (71/76)	94.3 (33/35)
R1	3.9 (8/203)	1.1 (1/92)	6.6 (5/76)	5.7 (2/35)
R2	0.5 (1/203)	1.1 (1/92)	.	.
Most common reasons for not performing SUR, among unresected patients (unknown for n = 20) ^‡^				
Poor cardiorespiratory reserve	31.9 (23/72)	38.9 (7/18)	22.2 (2/9)	31.1 (14/45)
Comorbidity	29.2 (21/72)	44.4 (8/18)	22.2 (2/9)	24.4 (11/45)
Advanced age	20.8 (15/72)	16.7 (3/18)	44.4 (4/9)	17.8 (8/45)
Unresectable disease stage IIIC	12.5 (9/72)	.	.	20.0 (9/45)
Patient frailty	11.1 (8/72)	5.6 (1/18)	22.2 (2/9)	11.1 (5/45)
ST (±RT or SUR)	43.6 (139/319)	7.8 (9/115)	43.9 (43/98)	82.1 (87/106)
Most common reasons for not receiving ST, among those not receiving ST (unknown for n = 28) ^‡^				
Not indicated	66.4 (93/140)	83.9 (73/87)	43.2 (19/44)	11.1 (1/9)
Patient’s refusal	9.3 (13/140)	1.1 (1/87)	20.5 (9/44)	33.3 (3/9)
Tumor board decision	7.9 (11/140)	9.2 (8/87)	2.3 (1/44)	22.2 (2/9)
RT (±ST or SUR)	22.3 (71/319)	16.5 (19/115)	14.3 (14/98)	35.8 (38/106)
CRT (±SUR)	10.0 (32/319)	.	1.0 (1/98)	29.2 (31/106)
cCRT, among those receiving CRT	50.0 (16/32)	.	100.0 (1/1)	48.4 (15/31)
sCRT, among those receiving CRT	46.9 (15/32)	.	.	48.4 (15/31)
Both cCRT and sCRT, among those receiving CRT	3.1 (1/32)	.	.	3.2 (1/31)
**Time from histological NSCLC confirmation to start of ITS, among patients receiving that treatment, median (IQR)** ** ^¶^ **				
Any (non)pharmacologic treatment, days ^§^	24.0 (7.0–39.0)	25.5 (0.0–40.0)	27.0 (7.0–42.0)	21.5 (12.0–33.0)
SUR, days ^§^	26.0 (0.0–44.0)	19.0 (0.0–34.0)	26.0 (2.0–44.0)	53.0 (21.0–101.0)
RT, months	2.1 (1.3–3.5)	1.4 (1.2–1.7)	1.9 (1.0–2.5)	2.9 (1.7–4.5)
ST, months	1.2 (0.7–2.0)	2.2 (1.9–2.4)	2.0 (1.4–2.8)	0.8 (0.5–1.3)
**Pharmacologic categories and drug classes among patients treated with ST, % (n/N)**				
Chemotherapy	94.2 (131/139)	100.0 (9/9)	97.7 (42/43)	92.0 (80/87)
Platinum compound	93.5 (130/139)	100.0 (9/9)	97.7 (42/43)	90.8 (79/87)
Folic acid analogue	39.6 (55/139)	66.7 (6/9)	37.2 (16/43)	37.9 (33/87)
Vinca alkaloid and analogue	26.6 (37/139)	22.2 (2/9)	46.5 (20/43)	17.2 (15/87)
Taxane	14.4 (20/139)	11.1 (1/9)	.	21.8 (19/87)
Antimetabolite	13.7 (19/139)	.	11.6 (5/43)	16.1 (14/87)
Topoisomerase II inhibitor	1.4 (2/139)	.	2.3 (1/43)	1.1 (1/87)
ICI	28.1 (39/139)	.	4.7 (2/43)	42.5 (37/87)
Anti-PD-1	20.9 (29/139)	.	4.7 (2/43)	31.0 (27/87)
Anti-PD-L1	7.2 (10/139)	.	.	11.5 (10/87)
Targeted therapy	0.7 (1/139)	.	.	1.1 (1/87)
ALK TKI	0.7 (1/139)	.	.	1.1 (1/87)

For variables not following a normal distribution in at least one of the study subpopulations, a uniform presentation of median (IQR) was applied. ^†^ Excluding supportive treatments; ^⌠^ Other types of resection were reported in ≤10 patients, each. ^∫^ Other surgical approaches were reported in ≤2 patients, each. ^‡^ Reported in ≥20.0% in any of the examined subpopulations; ^¶^ Imputation of the start of ITS has been implemented for 2 patients due to unknown day and/or month; ^§^ Nine patients have performed surgery before the histological confirmation of NSCLC. Abbreviations: ALK, Anaplastic Lymphoma Kinase; cCRT, concurrent CRT; CRT, Chemoradiation; ICI, Immune Checkpoint Inhibitor; IQR, Interquartile Range; ITS, Initial Therapeutic Strategy; N, number of patients with available data; n, number of patients with variable; NSCLC, Non-Small Cell Lung Cancer; PD-1, Programmed Cell Death 1; PD-L1, Programmed Death Ligand 1; RT, Radiotherapy; sCRT, sequential CRT; ST, Systemic Therapy; SUR, Surgery; TKI, Tyrosine Kinase Inhibitors; VATS, Video-Assisted Thoracoscopic Surgery.

## Data Availability

Data is not available due to ethical restrictions.

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
