# Peer review of "Real-World Treatment Patterns and Timeliness of Clinical Care Pathway for Non-Small Cell Lung Cancer Patients in Austria: The PRATER Retrospective Study"

_cancers, 2024, doi:10.3390/cancers16142586_

Round 1

Reviewer 1 Report

Comments and Suggestions for Authors

In this original manuscript, the authors conducted a PRATER retrospective study to characterize the profile, the different therapeutic strategies used, and the path of patients diagnosed with non-small cell lung cancer between the years 2018 and 2021. It was considered since the patients were diagnosed until the end of the patient's observation period, taking into account the earliest date of completion of the initial therapeutic strategy, the last contact of the patient, death, or completion of the study. The authors collected data from different Austrian hospitals to carry out the study. In addition, their results were divided according to the different stages of the disease. In addition, as a second stage of the study, the authors analyzed the impact that the World COVID-19 pandemic could have on the prevalence of new cases of non-small cell lung cancer between 2020 and 2021. The authors also divided the impact of this pandemic, taking into account the different stages of the disease. The work of this research group could be of great importance since they show a general visualization, especially of the most used techniques. However, I think it is limited to exposing the most useful therapeutic strategies according to mortality or other important parameters. The latter do not have enough importance within the study and from my point of view this would have to improve. However, I think this work is complete and written properly. The results are clearly explained. The conclusion is supported by the results. Therefore, it is suggested to accept this work after taking into account the following minor reviews:

-        Carefully check the abbreviations within the summary. For example, it is necessary to define KRAS and EGFR.

-        Why are the "N" sums not equal to the "N"? For example, in "Receipt of (NEO) AGUVANT ST" are 19/215 in "Receipt of neoadjuvant ST" and 63 of 215 in "Receipt of adjuvant ST." are there more types of ST (neo) adjuvant reception? Since 19 plus 63 are less than 50% of the population, which reception of ST (neo) adjuvant is more common? The same questions could be applied to the other parameters.

-        The authors are suggested to make an image of the methods and what were the measurements. This could help clarity for the reader.

Author Response

  1. Carefully check the abbreviations within the summary. For example, it is necessary to define KRAS and EGFR.

I would like to thank the reviewer for pointing it out. We have spelled out PD-L1, KRAS and EGFR in the revised manuscript Abstract (see lines 47-49).

  1. Why are the "N" sums not equal to the "N"? For example, in "Receipt of (NEO) AGUVANT ST" are 19/215 in "Receipt of neoadjuvant ST" and 63 of 215 in "Receipt of adjuvant ST." are there more types of ST (neo) adjuvant reception? Since 19 plus 63 are less than 50% of the population, which reception of ST (neo) adjuvant is more common? The same questions could be applied to the other parameters.

Regarding the question about (neo)adjuvant ST, in each case, n/N is presented in parentheses. For example, 67.4% received surgery (215/319), but receipt of (neo)adjuvant is shown as % among those receiving surgery, hence: 8.8% of resected patients (19/215) received neoadjuvant and 29.3% of resected patients (63/215) received adjuvant; thus, adjuvant is more common (5 resected patients received both neoadjuvant and adjuvant ST; see Figure 2). Further down Table 2: ST (regardless of whether it was primary/neoadjuvant/adjuvant/maintenance/consolidation) is shown as % among all patients: 43.6% of all patients (139/319) received ST. The explanation for differences seen in ST frequencies between stages is given in the respective text (lines 249-253).

We agree that this can be confusing in Table 2 since the denominator can be smaller either due to the subgroup being analyzed or due to missing data for a particular variable or both reasons (e.g. surgical approach). To address this comment, we have included explanatory text in the respective labels (highlighted in Table 2 of the revised manuscript).

  1. The authors are suggested to make an image of the methods and what were the measurements. This could help clarity for the reader.

Thank you for this very nice suggestion. A figure was added in the Supplementary file (see Figure S1 and revised manuscript line 116).

Reviewer 2 Report

Comments and Suggestions for Authors

  Retrospective studies are designed to analyse pre-existing data, and are subject to numerous biases as a result. For instance, this was a retrospective study of the profile and initial treatments of adults diagnosed 44 with early-stage non-small cell lung cancer (NSCLC) during Jan/2018-Dec/2021 at 16 leading 45 hospital institutions in Austria, excluding patients enrolled in clinical trials. In total, 319 patients 46 were enrolled at a planned 1:1:1 ratio across StI:II:III.

1. If this was done retrospectively, how were patients enrolled exactly 1:1:1?
There was selection bias toward the authors' intention since this study is
based on chart reviews (data collection from the medical records of patients)The proportion of stage I~III is never 1:1:1 in real-world data.

2. The authors chose an inappropriate acronym for immunotherapy (IO stands for immuno-oncology)

3. The references are outdated. The authors did not provide the latest information. For example. Reference 1 may be replaced by Global cancer statistics 2022: GLOBOCAN estimates of incidence and mortality worldwide for 36 cancers in 185 countries.Bray F, et al. CA Cancer J Clin. 2024. PMID: 38572751 As this study lacks the latest information, it’s misleading because the most recent studies might correct, refute, or build upon older research.

Author Response

  1. If this was done retrospectively, how were patients enrolled exactly 1:1:1?

There was selection bias toward the authors' intention since this study is based on chart reviews (data collection from the medical records of patients)The proportion of stage I~III is never 1:1:1 in real-world data.

Indeed, representing the real-world stage distribution was not the aim of the primary part of the study (based on the exploratory part, Stage I, II, III and IV was 22%, 10%, 25%, and 43%). Instead, a 1:1:1 ratio was implemented for the primary study part in order to achieve a meaningful number of patients in each study subpopulation and ensure a margin of error <10% for the frequency estimates (this is also mentioned in the ‘Statistical methods’ section). Additional information has been added in the revised manuscript (see lines 121 and 126-128).

  1. The authors chose an inappropriate acronym for immunotherapy (IO stands for immuno-oncology).

As there is some level of inconsistency in the literature regarding this abbreviation, to address this comment, IO has been changed from immunotherapy to immune checkpoint inhibitor (ICI) across all instances in the revised manuscript, including Table 2, Figure 2 and Figure 4.

  1. The references are outdated. The authors did not provide the latest information. For example. Reference 1 may be replaced by Global cancer statistics 2022: GLOBOCAN estimates of incidence and mortality worldwide for 36 cancers in 185 countries.Bray F, et al. CA Cancer J Clin. 2024. PMID: 38572751 As this study lacks the latest information, it’s misleading because the most recent studies might correct, refute, or build upon older research.

Thank you for notifying us of the new GLOBOCAN publication, which became available during the time elapsed between manuscript development and journal submission. The numbers and corresponding reference have been amended in the revised manuscript (see lines 65-66 and 635-637). Fortunately, the numbers did not change significantly (11% changed to 12% and 18% changed to 19%), thus having a negligible impact on the interpretation of this sentence.

With respect to the references further down, our aim was to give the picture of poor outcomes prior to ICIs and TT as it better corresponds to the PRATER index period (2018-2021). Hopefully it is clearer now, after rephrasing the sentence in lines 72-75, in combination with the timeline of new indications as presented in lines 90-101. In addition, first appearance of the reference Houda et al., 2024 (timeline of new indications) has been moved up to line 74.

Reviewer 3 Report

Comments and Suggestions for Authors

Thank you for the opportunity to review your manuscript titled "Real-world treatment patterns and timeliness of clinical care pathway for non-small cell lung cancer patients in Austria: the PRATER retrospective study." Your research provides valuable insights into the treatment patterns and clinical care of NSCLC patients in Austria, particularly during the COVID-19 pandemic. Below are my comments and suggestions for improving the manuscript:

Major Comments

1. Detailed Examination of Clinical Staging:

 Please consider a more detailed examination of clinical staging. This is particularly important for stages IA and IB, as well as IIIA and IIIB, where there are differences in the indications for adjuvant chemotherapy and the choice between surgical treatment and chemoradiotherapy.

2. Specification of Detection Methods for Driver mutation and PD-L1:

 The detection methods for driver gene alterations and PD-L1 expression significantly influence the results. To enhance the clarity and reliability of the findings, it is essential to explicitly specify the detection methods used.

Author Response

  1. Detailed Examination of Clinical Staging:

 Please consider a more detailed examination of clinical staging. This is particularly important for stages IA and IB, as well as IIIA and IIIB, where there are differences in the indications for adjuvant chemotherapy and the choice between surgical treatment and chemoradiotherapy.

We agree that differences in treatment patterns may exist between substages. However, an examination of the study endpoints among subgroups by substage was beyond the scope of this study due to the underlying complexity. As this analysis is not currently available, lack thereof is discussed in the revised manuscript as part of the study’s limitations (see lines 567-570).

  1. Specification of Detection Methods for Driver mutation and PD-L1:

 The detection methods for driver gene alterations and PD-L1 expression significantly influence the results. To enhance the clarity and reliability of the findings, it is essential to explicitly specify the detection methods used.

Thank you for your suggestion. To address this comment we have included all details in a new Supplementary Table (Table S1; revised manuscript lines 226-228), since the frequency of each method (NGS/IHC/other) was different for each biomarker, and thus would complicate the main figure. Kindly note that with respect to PD-L1, IHC-based results are presented [see legend of Figure 1A ‘PD-L1 TPS via IHC’ and manuscript lines 228-229 ‘PD-L1 (TPS ≥1% via immunohistochemistry; 57.9%) (Figure 1A)’].

Reviewer 4 Report

Comments and Suggestions for Authors

I read with interest the research article entitled “Real-world treatment patterns and timeliness of clinical care 2 pathway for non-small cell lung cancer patients in Austria: the 3 PRATER retrospective study”. Hochmair and colleagues report a real-world experience by a multicentre observational retrospective study in Austria, namely PRATER study, to evaluate the clinical management performance for early stage (I-III) NSCLC patients.

This reviewer only has a few suggestions:

- In the study design, please add which staging system has been adopted to stage patients, considering a 4-year observational period and the participation of 14 centres to electronic surveys, please add comments whether different staging systems (in time and centres) have been used.

- Table 1 is full of unnecessary data; it has to be simplified to be more readable. For age, it is sufficient to insert median and IQR range as I-III IQR, it makes no sense to display median and single IQR, the same is for BMI index. Moreover, in the whole table the partials don’t reflect the sum of the subgroups (overall, stage I, II, III); one can imagine that these data are missing. If so, just add a line for missing or N/A data, displaying percentages on the considered populations, without repeating n/N for each cell of the table (e.g. primary histology: non-squamous 50.6 (159); squamous 35.4 (111); missing 5.

- PD-L1 expression was categorized as ≥10.0% along the text and as 1-49% and ≥50.0% in Figure 1A. Please provide a uniform categorization.

- Figure 2 contains redundant information. It is unnecessary to include two different layers of the pie-chart to specify type of treatment (e.g. ChT and PDC in two different layers. It could be necessary to distinct ChT by other treatments , without specifying in the figure this; more information could be added in the text.

- Along the results it is often stated IQR, without specification about which quartiles are considered. I suggest to use first and third quartiles as range to express data and IQR; anyway, the selected quartiles for data expression need to be declared.

- The sub-sections of Discussion paragraph are questionable. I suggest the authors to make a single paragraph for the Discussion, or at least to reduce sub-sections.

Minor concerns:

- Lines 71-73: the sentence is unclear, it could be re-phrased indicating as immunotherapy and targeted therapy prolonged time-to-relapse and overall survival in early stage NSCLC. Reported references are appropriated.

Comments on the Quality of English Language

Minor spell check

Author Response

  1. In the study design, please add which staging system has been adopted to stage patients, considering a 4-year observational period and the participation of 14 centres to electronic surveys, please add comments whether different staging systems (in time and centres) have been used.

This information is not available since the exploratory part did not collect patient-level data, as per study design. To address this comment, relevant text has been added in the limitations section in the revised manuscript (see lines 556-560).

  1. Table 1 is full of unnecessary data; it has to be simplified to be more readable. For age, it is sufficient to insert median and IQR range as I-III IQR, it makes no sense to display median and single IQR, the same is for BMI index. Moreover, in the whole table the partials don’t reflect the sum of the subgroups (overall, stage I, II, III); one can imagine that these data are missing. If so, just add a line for missing or N/A data, displaying percentages on the considered populations, without repeating n/N for each cell of the table (e.g. primary histology: non-squamous 50.6 (159); squamous 35.4 (111); missing 5.

With respect to the IQR comment, please note that the IQR has been defined in line 173 of the revised manuscript (see also response to comment #5).

To simplify the table, we have removed information on race, retirement, current smoking status and comorbidities. Although we agree that the suggested format of ‘non-squamous 50.6 (159)’ is easier for the reader to follow, we did not proceed with this change. The reason being that it would also affect the format of Table 2, which contains further subgrouping, not only driven by missing data.

With respect to the sum of subgroups, a footnote was added (‡), where applicable, to help the reader understand that the denominator is smaller because of missing data (see lines 212-214). Please note that the next new footnote (⌠) explains why the percentages don’t add up to 100% (see lines 214-215).

  1. PD-L1 expression was categorized as ≥10.0% along the text and as 1-49% and ≥50.0% in Figure 1A. Please provide a uniform categorization.

Please note that ≥10.0% referred to biomarkers being positive in ≥10.0% of patients, not the PD-L1 cut-off; 57.9% refers to those expressing PD-L1 (see Figure 1A: 24.0%+33.1%+0.8%). To address this comment and avoid any misinterpretation, we have deleted ‘(≥10.0%)’ since it is obvious from the figure that these biomarkers have the highest positivity. In addition, we added ‘TPS ≥1%’ in parentheses for PD-L1 in the revised manuscript (see line 228).

  1. Figure 2 contains redundant information. It is unnecessary to include two different layers of the pie-chart to specify type of treatment (e.g. ChT and PDC in two different layers. It could be necessary to distinct ChT by other treatments , without specifying in the figure this; more information could be added in the text.

We agree and thank the reviewer for guiding us towards making this figure simpler to the reader. A new figure has been added which still includes all the information but as 2 layers instead of 3.

  1. Along the results it is often stated IQR, without specification about which quartiles are considered. I suggest to use first and third quartiles as range to express data and IQR; anyway, the selected quartiles for data expression need to be declared.

We sincerely appreciate pointing out this important omission. When stating IQR we are in fact referring to the first and third quartiles throughout the manuscript. This has now been defined in the revised manuscript line 173.

  1. The sub-sections of Discussion paragraph are questionable. I suggest the authors to make a single paragraph for the Discussion, or at least to reduce sub-sections..

The subsections have been reduced to 3 simple sections mirroring the ‘Results’ section (see lines 427, 474-475, 493, and 542 of the revised manuscript).

  1. Lines 71-73: the sentence is unclear, it could be re-phrased indicating as immunotherapy and targeted therapy prolonged time-to-relapse and overall survival in early stage NSCLC. Reported references are appropriated.

The sentence has been rephrased (see lines 72-75 of the revised manuscript), which hopefully makes it clear now that the point we are trying to make is on the poor outcomes which however are not reflective of the current situation/today which might have improved owing to the new indications of ICIs and TTs.

Round 2

Reviewer 3 Report

Comments and Suggestions for Authors

After careful review, your manuscript entitled "Real-world treatment patterns and timeliness of clinical care pathway for non-small cell lung cancer patients in Austria: the PRATER retrospective study" (Manuscript ID: cancers-3077650) has met the standards for acceptance.

Your research offers significant insights into the real-world treatment patterns and clinical pathways for non-small cell lung cancer patients in Austria, particularly highlighting the timeliness of care and adherence to guidelines. The comprehensive nature of your study, encompassing data from 16 leading hospital institutions, provides a robust analysis that will be valuable to the oncology community.

The revisions you have made in response to earlier feedback have strengthened the manuscript, and the clarity and thoroughness of your presentation are commendable. Your findings contribute important knowledge to the field, especially concerning treatment practices prior to the era of immunotherapy and targeted therapies in the (neo)adjuvant setting.